# Impact of continuous labor companion- who is the best: A systematic review and meta-analysis of randomized controlled trials

D. M. C. S. Jayasundara[1,2]*, I. A. Jayawardane[1,2☯], S. D. S. Weliange[3☯], T. D. K. M. Jayasingha[1☯], T. M. S. S. B. Madugalle[4☯]

1 Department of Obstetrics and Gynaecology, Faculty of Medicine, University of Colombo, Colombo, Sri Lanka, 2 De Soysa Maternity Hospital, Colombo, Sri Lanka, 3 Department of Community Medicine, Faculty of Medicine, University of Colombo, Colombo, Sri Lanka, 4 National Hospital, Kandy, Sri Lanka

☯ These authors contributed equally to this work.
* chandana@obg.cmb.ac.lk

## Abstract

**Data Availability Statement:** All relevant data are within the manuscript and its Supporting Information files.

### Background

Continuous labor support is widely acknowledged for potentially enhancing maternal and neonatal outcomes, the physiological labor process, and maternal satisfaction with the labor experience. However, the existing literature lacks a comprehensive analysis of the optimal characteristics of labor companions, particularly in comparing the effects of trained versus untrained and familiar versus unfamiliar labor companions across diverse geographical regions, both pre-and post-millennial. This meta-analysis addresses these research gaps by providing insights into the most influential aspects of continuous labor support.

### Methodology

A thorough search of PubMed, Google Scholar, Science Direct, International Clinical Trials Registry Platform (ICTRP), ClinicalTrials.gov, Research4Life, and Cochrane Library was conducted from 25/06/2023 to 04/07/2023. Study selection utilized the semi-automated tool Rayyan. The original version of the Cochrane Risk of Bias tool was used to assess the quality of Randomized Controlled Trials (RCTs) while funnel plots gauged the publication bias. Statistical analysis employed RevMan 5.4, using Mantel-Haenszel statistics and random effects models to calculate risk ratios with 95% confidence intervals. Subgroup analyses were performed for different characteristics, including familiarity, training, temporal associations, and geographical locations. The study was registered in INPLASY (Registration number: INPLASY202410003).

### Results

Thirty-five RCTs were identified from 5,346 studies. The meta-analysis highlighted significant positive effects of continuous labor support across various outcomes. The highest overall effect without subgroup divisions was the improvement reported in the 5-minute Apgar

**Funding:** The author(s) received no specific funding for this work.

**Competing interests:** The authors have declared that no competing interests exist.

score < 7, with an effect size of 1.52 (95% Confidence Interval (CI) 1.05, 2.20). Familiar labor companions were better at reducing tocophobia, with an effect size of 1.73 (95% CI 1.49, 2.42), compared to unfamiliar companions, with an effect size of 1.34 (95% CI 1.14, 1.58). Untrained labor companions were the better choice in reducing tocophobia and the cesarean section rate compared to trained companions. For the analysis of tocophobia, the pooled effect sizes were 1.34 (95% CI 1.14, 1.57) and 1.84(95% CI 1.60, 2.12) in trained versus untrained subgroup comparisons. For the cesarean rate, they were represented as 1.22 (95% CI 1.05, 1.42) and 2.16 (95% CI 1.37, 3.40), respectively. The pooled effect size for the duration of labor was 0.16 (95% CI 0.06, 0.26) for the subgroup of RCTs conducted before 2000 and 0.53 (95% CI 0.30, 0.77) for the subgroup of RCTs conducted after 2000. A significant subgroup difference (<0.1) was found in relation to the duration of labor, cesarean section rate, oxytocin for labor induction, analgesic usage, and tocophobia in the subgroup analysis of geographical regions.

## Discussion and conclusion

The beneficial effects of a labor companion are well-established in the literature. However, studies systematically assessing the characteristics of labor companions for optimal beneficial effects are lacking. The current study provides insights into the familiarity, training, temporal association, and geographical settings of labor companions, highlighting the differing impact of these characteristics on measured outcomes by evaluating the current randomized controlled trials on the topic. There is insufficient evidence to define the 'best labor companion' owing to the heterogeneity of labor companions and outcome assessment across different studies. We encourage well-designed further research to fill the research gap.

## Introduction

### Background

The emotional process of labor and childbirth is often a fearful and stressful event for a pregnant mother [1]. The term 'tocophobia,' originating from the Greek 'tokos' (childbirth) and 'phabos' (fear and anxiety), is an extreme anxiety or fear of childbirth [2]. In most cultures, the tradition of supporting a woman in labor is a community event with multiple participants other than the designated healthcare provider. The fear and anxiety of childbirth are augmented by an unfamiliar hospital environment, medical jargon, procedures, interventions, and transient separation from the family during labor [3]. The woman feels a sense of loss of control, isolation, and fear, peaking the level of anxiety [3]. To cope with this tocophobia, pregnant women sometimes choose cesarean section over natural birth [4]. Increased anxiety makes the woman more vulnerable to increased pain perception, prolonging the duration of labor and contributing to dystocia [5]. The pain and anxiety during labor increase the endogenous catecholamine release, causing ineffective uterine contractions and decreased placental blood flow [6]. An inefficient labor process may cause fetal and maternal complications, including the risk of fetal or neonatal hypoxia and death, infection, physical damage in the newborn, postpartum hemorrhage, maternal infection, and psychological distress due to anxiety, lack of sleep, and fatigue [5].

## Strategies to alleviate tocophobia

Different clinical settings have adopted strategies to alleviate tocophobia, facilitating a smooth labor process. Support methods include having an accompanying companion for continuous labor support, induced sleep, hydrotherapy, and the Lamaze relaxation method [7]. The labor companion can be a non-caregiving nurse, midwife, friend, relative, family member, husband, or a person trained in supporting labor (doula) [4]. WHO defines labor support as the supportive care provided to women during labor, including emotional support, physical comfort, advice, and information giving [6]. WHO also recommends that a parturient should have a birth companion of her choice. However, it is not practiced in many lower-middle-income counties (LMIC) [8].

## Beneficial effects of a labor companion

A companion for continuous labor support facilitates a smooth labor process, improving the maternal psychological status and fetal/neonatal well-being. Reported advantages include an increase in spontaneous vaginal births, reduced demand for analgesics, reduced need for oxytocin for labor augmentation, shorter duration of labor, decreased need for cesarean sections, minimal perineal trauma, and reduced requirement for instrumentation during labor, facilitating a smooth labor process [9–12]. Maternal psychological well-being is improved by lowering tocophobia, reduced postpartum depression and anxiety, and improved self-esteem and satisfaction measured postpartum [3, 13, 14]. Fetal/neonatal well-being is enhanced by the early establishment of exclusive breastfeeding, early skin-to-skin contact, and reduced need for neonatal resuscitation and neonatal hospital stay [15, 16].

## Gap of knowledge

The quality of labor support and its beneficial outcomes depend on the type of companion used [17]. The labor companion can be trained or untrained and familiar or unfamiliar to the parturient. The evidence regarding the "best labor companion" is controversial, and studies do not show a clear consensus.

The rates of severe tocophobia, measured similarly, vary in different countries, and the reasons are unknown [18]. The prevalence of tocophobia was lower in the early years (1980s, 1990s) compared to more recent years (2000 onwards) [19]. The beneficial effects of a labor companion can be more pronounced in some countries than others and may have changed over time.

## Objectives

The present meta-analysis aims to describe the characteristics of the most effective labor companion, highlighting the differences in beneficial effects of having a labor companion among different geographical regions and timelines.

# Materials and methods

## Search strategy

PubMed, Science Direct, Cochrane Library, Google Scholar, ClinicalTrials.gov, and International Clinical Trials Registry Platform (ICTRP) were searched from 25/06/2023 to 04/07/2023. To identify relevant studies, a set search strings such as "Labor companion," "Birth partner," "Doula," "Labor support person," "Childbirth coach," "Labor assistant," "Labor coach," "Birth attendant," "Labor caregiver," "Maternity support person," "Childbirth companion," "Labor ally," "Labor chaperone," "Pregnancy outcome," "Obstetric outcome," "Delivery

outcome," "Birth outcome," "Fetal outcome," "Newborn outcome," "Infant outcome," "Neonatal outcome", and "Baby's outcome" were employed, with Boolean expressions "AND" and "OR" used appropriately to construct precise search queries. Initially, the literature search was conducted without filters. Then, the results were refined using advanced search options like full-text articles and randomized controlled trials.

A manual search strategy was also applied to ensure inclusivity, focusing on identifying any missing studies by reviewing the most cited ten meta-analyses within the same databases.

## Screening eligible studies

The study selection process was carried out meticulously in two rounds using a semi-automated tool, Rayyan [20], with one author as the reviewer (DMCSJ) and another as a collaborator (TDKMJ), employing a blind approach. In the first round, titles and abstracts were screened, eliminating duplicates and ineligible entries, with conflicts resolved by the reviewer (IAJ). The second round involved a similar blind approach for full-text screening, again with conflicts resolved by the reviewer (IAJ). The authors were contacted if additional information was required. The study selection process was transparently reported using the PRISMA 2020 flow diagram for updated systematic reviews [21].

A detailed search strategy is given as a separate file under supporting information (S10 File). A protocol exists for the current study, and a copy of the protocol is given as supporting information (S11 File).

## Inclusion and exclusion criteria

Randomized controlled trials (RCTs) with full-text articles reporting results related to low-risk women with viable singleton pregnancies in cephalic presentation, admitted during the latent phase (cervical dilation 3–4 cm) with no contraindications for vaginal delivery were included in the study. RCTs in English language only were included. Studies reporting women with medical or psychiatric diseases, previous cesarean section, pelvic abnormalities not favoring vaginal birth, fetal distress, and any fetal anomaly were excluded. Any study designs other than RCT, including quasi-experimental trials, were excluded.

## Data extraction

Key study characteristics were extracted and organized into predefined tables for outcome measures concerning facilitating the labor process, maternal psychological well-being, and fetal well-being. To ensure the integrity of the research, a second author independently reviewed the entire process, minimizing the potential for bias.

## Risk of bias and quality assessment

The quality of each RCT was assessed using the original version of the Cochrane Risk of Bias tool [22]. Random sequence generation, allocation concealment, performance bias, detection bias, attrition bias, reporting bias, and other biases are used as criteria in the original version of the Cochrane Risk of Bias tool. Funnel plots were employed to gauge publication bias, with any deviation from the expected funnel-shaped distribution as an indicator of potential publication bias.

## Primary outcomes

Primary outcome measures were the overall effectiveness of a labor companion regarding spontaneous vaginal delivery, analgesic usage, oxytocin for labor induction, duration of labor,

in labor cesarean section, instrumental vaginal delivery, 5-min Apgar score, and tocophobia, without subgroup divisions.

## Secondary outcomes

Secondary outcomes compared the effects of trained versus untrained labor companions, familiar versus unfamiliar labor companions, studies before versus after 2000, and studies in different geographical locations by introducing subgroups.

## Description of subgroups

Subgroups were trained versus untrained labor companions, familiar versus unfamiliar labor companions, studies before versus after 2000, and studies in different geographical locations. Trained labor companions had one or more training sessions for the role of a labor companion before childbirth. In contrast, untrained labor companion directly attended childbirth without prior training about their expected role. A familiar labor companion was chosen by laboring women, usually a family member, friend, relative, or partner. In contrast, the healthcare facility introduced an unfamiliar labor companion, usually a community member or staff. Geographical locations were divided on a continental basis [23].

## Statistical analysis

We used RevMan version 5.4 to analyze the following outcome measures reported by more than ten RCTs—spontaneous vaginal birth, tocophobia, use of analgesics, need for synthetic oxytocin, duration of labor, CS rate, instrumental vaginal delivery, and 5-min Apgar score. The Mantel-Haenszel statistical method, random effects analysis model, and risk ratio with a 95% confidence interval (CI) as effect measures were used for dichotomous data. For the continuous data inverse variance statistical method, the random effects analysis model and standard mean difference as effect measures were used. We assessed heterogeneity with the $I^2$ statistic, considering p value $< 0.1$ or $I^2 > 50\%$ indicators of significant heterogeneity. Subgroup analyses compared the effects of trained versus untrained labor companions, familiar versus unfamiliar labor companions, studies before versus after 2000, and studies in different geographical locations.

## Results

### Search results, study characteristics, and quality assessment

Fig 1 shows the PRISMA 2020 flow diagram for study selection. We identified 5346 studies from 7 databases and a manual search. After considering exclusion and inclusion criteria, 35 studies were selected for analysis.

Table 1 summarizes the key characteristics of 35 RCTs, including the year of the study, country, number of participants, a description of the type of labor companion, and outcome measures. Studies span from 1986 to 2022 from various geographical regions: Asia, Africa, Europe, North America, South America and Australia. Three studies (8.57%) have less than 100 participants, while 7 (20%) have more than 500 participants. Hodnett (2002) from the USA had the highest number of participants, at 6915. Different studies have used labor companions with varying characteristics, such as familiar, unfamiliar, trained, and untrained. Twenty-three studies (65.71%) used trained labor companions, while 20 (57.14%) used unfamiliar ones. The individual studies have examined different outcome measures, with a recent emphasis on maternal psychological well-being.

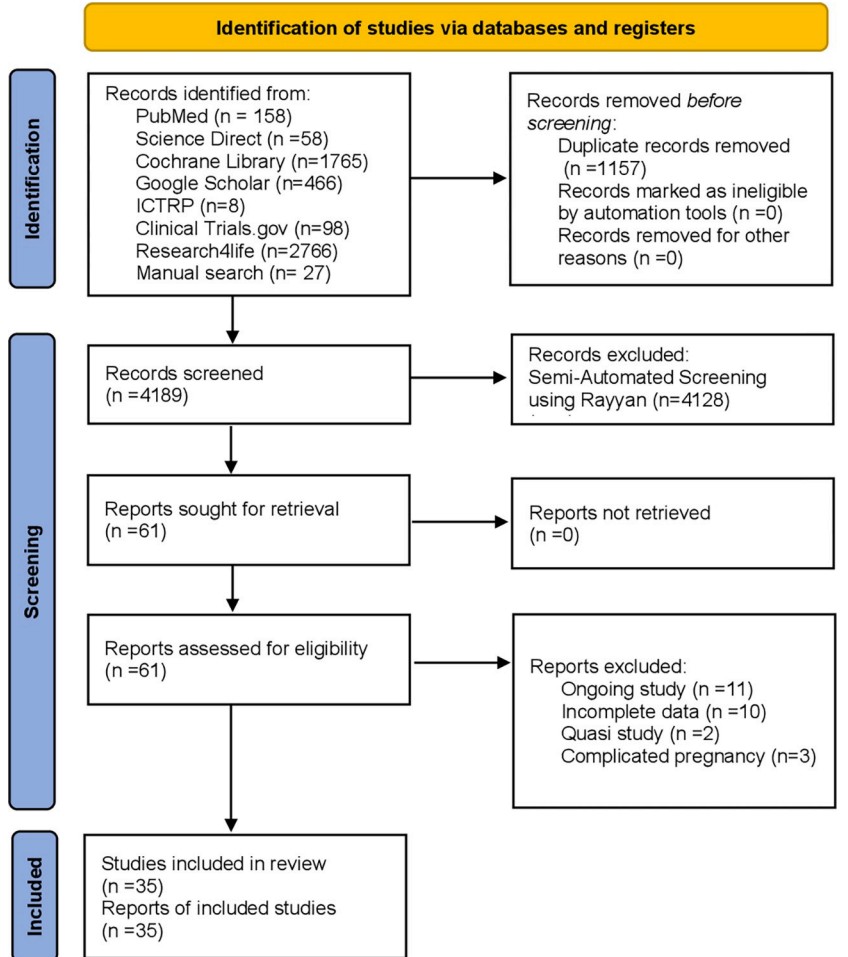

**Fig 1. PRISMA 2020 flow diagram for study selection after initial filtering.**

The funnel plots for each primary meta-analysis exhibited symmetry, suggesting minimal publication bias. Figs 2 and 3 summarize the original version of the Cochrane Risk of Bias tool assessment of RCTs. All the studies show an overall high risk of bias due to having unclear risk for multiple domains or a high risk of bias in at least one domain [43]. As expected, the highest risk of bias is reported in the blinding of participants and personnel (Fig 3).

## Primary analysis

Table 2 reports a meta-analysis of 8 outcomes as risk ratios and standard mean differences with 95% confidence intervals. The highest overall effect is the improvement reported in the 5 min Apgar score < 7 by 1.52(95% CI 1.05,2.20). All the outcomes are statistically significant, but most show a moderate to low effect size. Considerable heterogeneity is also reported in all results except for 5-minute Apgar < 7 and instrumental delivery. Fig 4 displays the forest plot for the meta-analysis of the 5-minute Apgar score.

## Secondary analysis

**Familiar versus unfamiliar labor companion.** Table 3 shows the subgroup analysis of the same eight outcomes shown in Table 2 for familiar versus unfamiliar labor companions.

**Table 1. Key characteristics of randomized controlled trials analyzed in the meta-analysis.**

| Study | Year | Country | Participants | Labor companion | Primary outcomes compared to the control group |
|---|---|---|---|---|---|
| Akbarzadeh [24] | 2014 | Iran | 100 | Researcher as doula | Increased maternal satisfaction, Reduced fetal distress |
| Bello [17] | 2009 | Nigeria | 585 | Untrained familiar companion | Reduction in CS, pain scores, and duration of labor. Improved maternal satisfaction. |
| Bolbol [5] | 2016 | Iran | 100 | Unfamiliar midwifery students | Improved 5-min Apgar score, Reduced Duration of labor. |
| Breat [25] | 1992 | Belgium | 262 | Unfamiliar midwives | Reduction in operative vaginal deliveries |
| Breat [25] | 1992 | France | 1319 | Unfamiliar midwives | Reduction in operative vaginal deliveries |
| Breat [25] | 1992 | Greece | 545 | Unfamiliar midwives | No difference in operative vaginal deliveries |
| Bruggemann [9] | 2007 | Brazil | 212 | Untrained familiar companion | Decreased incidence of meconium-stained amniotic fluid. Improved maternal satisfaction. |
| Campbell [26] | 2007 | USA | 494 | Trained familiar community doula | Improved maternal satisfaction and self-esteem. |
| Campbell [10] | 2006 | USA | 586 | Trained familiar community doula | Decreased duration of labor, Improved 5-min Apgar score, No difference in intrapartum analgesic usage and CS. |
| Cogan [27] | 1988 | USA | 25 | Trained unfamiliar community doula | Shorter duration of labor, Reduced need for intrapartum analgesia, Improved neonatal well-being. |
| Dickinson [28] | 2002 | Australia | 992 | Unfamiliar midwives | Maternal satisfaction with epidural analgesia was higher compared to techniques like continuous labor companion. |
| Erica [16] | 2022 | Sweden | 143 | Trained familiar community doula | No difference in the rating of labor care and emotional well-being. |
| Gagnon [29] | 1997 | Canada | 100 | Unfamiliar nursing officers | Reduction in CS |
| Hemminki a [30] | 1990 | Finland | 122 | Unfamiliar midwifery students | Increased maternal satisfaction, progress in labor and interventions, and mother and infant health were similar in the two groups. |
| Hemminki b [30] | 1990 | Finland | 118 | Unfamiliar midwifery students | Increased maternal satisfaction, progress in labor and interventions, and mother and infant health were similar in the two groups. |
| Hodnett [31] | 1989 | Canada | 103 | Trained familiar birth attendants | Less intrapartum analgesia, No difference in Duration of labor and CS, Increased use of Oxytocin |
| Hodnett [11] | 2002 | USA | 6915 | Unfamiliar nursing officers | No difference in CS, maternal or neonatal events during labor, and hospital stay. |
| Hofmeyr [32] | 1991 | South Africa | 189 | Untrained, unfamiliar community doula | Lower pain and anxiety scores, No measurable effect on the progress of labor |
| Isbir [13] | 2015 | Turkey | 72 | Unfamiliar midwifery students | Less fear during delivery, lower pain scores, and shorter delivery period. No difference in oxytocin use. |
| Kashanian [33] | 2010 | Iran | 100 | Unfamiliar midwives | Reduced Duration of labor and CS. Rates of Oxytocin use and 5-min Apgar scores were similar in both groups. |
| Kennell [34] | 1991 | USA | 412 | Trained unfamiliar community doula | Reduction in CS, instrumentation, analgesic usage, oxytocin use, duration of labor, and infant hospitalization. |
| Klaus [35] | 1986 | Guatemala | 417 | Unfamiliar, untrained community doula | Reduction in duration of labor, CS, oxytocin augmentation, and NICU admissions. |
| Langer [36] | 1998 | Mexico | 710 | Unfamiliar retired nursing officers | Reduction in duration of labor, No effects on medical interventions, maternal psychology, or newborn's condition |
| Madi [37] | 1999 | Botswana | 109 | Untrained female relative | Increase in spontaneous vaginal delivery, Less intrapartum analgesia, Fewer instrumentations, and CS, Less oxytocin. |
| McGrath [12] | 2008 | USA | 420 | Unfamiliar trained doula | Reduction in CS, analgesic usage, and positive experience with doula |
| Nikodem [38] | 1998 | South Africa | 39 | Unfamiliar, untrained community doula | No differences in postpartum depression and Coppersmith self-esteem scores. |
| Robati [4] | 2020 | Iran | 80 | Unfamiliar midwives | Decreased anxiety and pain during labor |
| Safarzadeh [6] | 2012 | Iran | 150 | Untrained friend/relative | Reduced duration of labor, No difference in pain severity. |
| Salehi [7] | 2016 | Iran | 84 | Trained husband/friend/relative | Reduced maternal anxiety |
| Shahshahan [8] | 2012 | Iran | 50 | Familiar, untrained support person | Reduction in labor and labor pain duration, Increased maternal satisfaction, and No difference in 5-min Apgar score and instrumentation. |

*(Continued)*

**Table 1.** (Continued)

| Study | Year | Country | Participants | Labor companion | Primary outcomes compared to the control group |
|---|---|---|---|---|---|
| Torres [39] | 1999 | Chile | 435 | Trained familiar companion | No difference in spontaneous vaginal deliveries, intrapartum analgesic use, oxytocin use, cesarean birth, and instrumentation. |
| Trotter [40] | 1992 | South Africa | 63 | Unfamiliar, untrained community doula | Reduced incidence of postpartum depression |
| Trueba [41] | 2000 | Mexico | 100 | Trained unfamiliar doula | Reduction in CS, Duration of labor, and analgesic usage. |
| Wolman [42] | 1993 | South Africa | 149 | Unfamiliar, untrained community doula | Increased self-esteem, Decreased postpartum depression and anxiety. |
| Yeonyong [3] | 2012 | Thailand | 114 | Trained relative | Shorter duration of labor, Increased maternal satisfaction, No difference in spontaneous vaginal deliveries. |

CS–Cesarean Section, NICU–Neonatal Intensive Care Unit

Subgroups based on trial or patient characteristics that modified outcomes were considered for statistical significance. P value from the test for subgroup difference was used to assess statistically significant subgroup differences. P<0.1 indicates a statistically significant subgroup difference [44]. There was no difference in outcomes in the subgroup analysis except for tocophobia (p = 0.02). Therefore, having a familiar labor companion reduced Tocophobia significantly. There was no significant subgroup heterogeneity concerning tocophobia within either subgroup. The combined effect size for Tocophobia was 1.73 (95% CI 1.49,2.02) for the familiar labor companion subgroup and 1.34 (95% CI 1.14,1.58) for the unfamiliar labor companion subgroup. However, there was an unequal distribution of trials and participants between the familiar and unfamiliar companion subgroups in all subgroup analyses. Nevertheless, for all eight outcomes, the pooled effect size for each subgroup favored having a continuous labor companion. Fig 5 displays the forest plot for the subgroup analysis of tocophobia.

### Trained versus untrained labor companion

We analyzed trained versus untrained labor companions in a subgroup analysis. The pooled effect size, irrespective of companion training, favored the presence of a labor companion for all outcomes. Only two outcomes, namely tocophobia (p = 0.004) and cesarean section (p = 0.02), differed between the subgroups. These findings suggest that companion training significantly negatively impacts these outcomes, which is difficult to explain. However, subgroup heterogeneity concerning tocophobia was significant within the trained companion group ($I^2$ = 59%) but not in the untrained companion group ($I^2$ = 0%). Conversely, subgroup heterogeneity was insignificant within the trained companion group ($I^2$ = 44%) but significant within the untrained companion group ($I^2$ = 54%) in relation to the CS rate. For the analysis of Tocophobia, the pooled effect sizes were denoted as 1.34 (95% CI 1.14,1.57) and 1.84 (95% CI 1.60,2.12) in the trained versus untrained subgroup comparisons. At the same time, the CS rate was represented at 1.22 (95% CI 1.05, 1.42) and 2.16 (95% CI 1.37, 3.40), respectively. There was a notable imbalance in the distribution of trials and participants between the trained and untrained companion subgroups across all eight outcomes (S1 Table).

### Effectiveness of labor companion before and after 2000

Out of the eight outcomes analyzed, a statistically significant subgroup difference was observed only in the duration of labor (p = 0.004), suggesting that the classification of RCTs as before and after 2000 has a significant impact. Notably, subgroup heterogeneity was significant within the subgroup of RCTs conducted after 2000 ($I^2$ = 74%) but not in the subgroup of RCTs

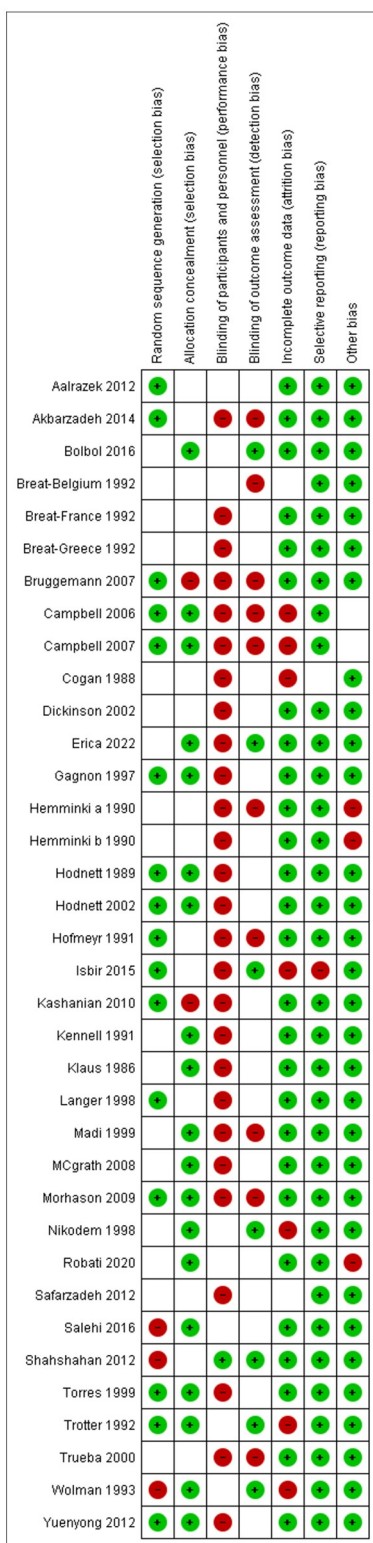

**Fig 2. Risk of bias summary.** Green: Low risk, Red: High risk, Blank: Unclear risk.

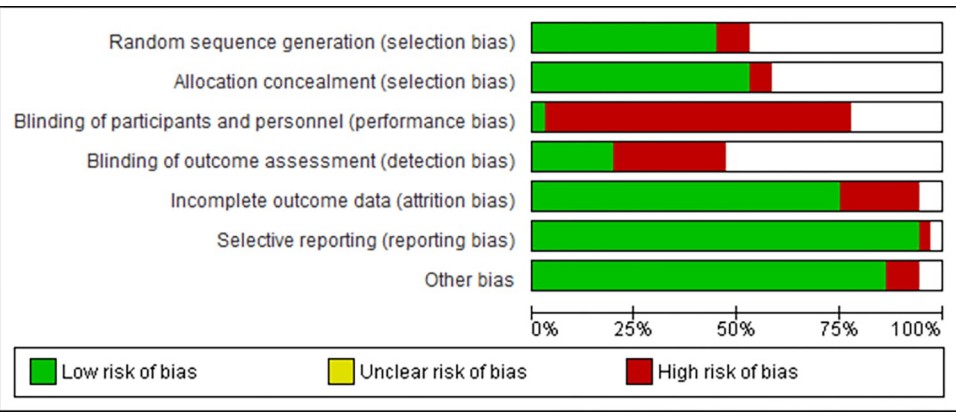

**Fig 3. Risk of bias graph.**

conducted before 2000 ($I^2$ = 44%) concerning the duration of labor. The pooled effect size for the duration of labor was 0.16 (95% CI 0.06,0.26) for the subgroup of RCTs conducted before 2000 and 0.53 (95% CI 0.30,0.77) for the subgroup of RCTs conducted after 2000. However, there was an uneven distribution of trials and participants between subgroups in all subgroup analyses. Nevertheless, having a continuous labor companion consistently showed a favorable pooled effect size in all eight outcomes (S2 Table).

## Effectiveness of labor companions in different geographical regions

Table 4 summarizes the effect of having a labor companion in different geographical regions such as Asia, Africa, and Europe. A significant subgroup difference (p<0.1) was found in relation to the duration of labor, CS rate, oxytocin for labor induction, analgesic usage, and tocophobia, indicating that ethnic differences significantly influence these outcomes. The most significant overall effects are seen in Asia, followed by Africa. Effects are minimal in the

**Table 2. Effectiveness of a labor companion related to 8 outcomes.**

| Outcome | No. of Participants (Studies) | RR (95% CI) | P value | Heterogeneity ($I^2$) |
|---|---|---|---|---|
| 1. Spontaneous vaginal delivery | 13811 (19 RCTs) | 1.09(1.04,1.13) | 0.0001 | 64% |
| 2. Duration of labor (Hours) (Standard mean difference) | 5422 (17 RCTs) | 0.30(0.18,0.41) | 0.0001 | 72% |
| 3. Cesarean section | 15080 (24 RCTs) | 1.43(1.20,1.71) | 0.0001 | 63% |
| 4. Instrumental delivery | 13955 (21 RCTs) | 1.13(1.03,1.23) | 0.008 | 23% |
| 5. Oxytocin for labor induction | 12958 (21 RCTs) | 1.10(1.01,1.19) | 0.03 | 71% |
| 6. Analgesic usage | 12719 (18 RCTs) | 1.06(1.01,1.11) | 0.02 | 52% |
| 7. Tocophobia | 11133 (11 RCTs) | 1.46(1.26,1.68) | 0.0001 | 63% |
| 8. 5 min Apgar < 7 | 12539 (16 RCTs) | 1.52(1.05,2.20) | 0.03 | 12% |

RCT–Randomized Controlled Trial, RR–Relative Risk, CI–Confidence Interval

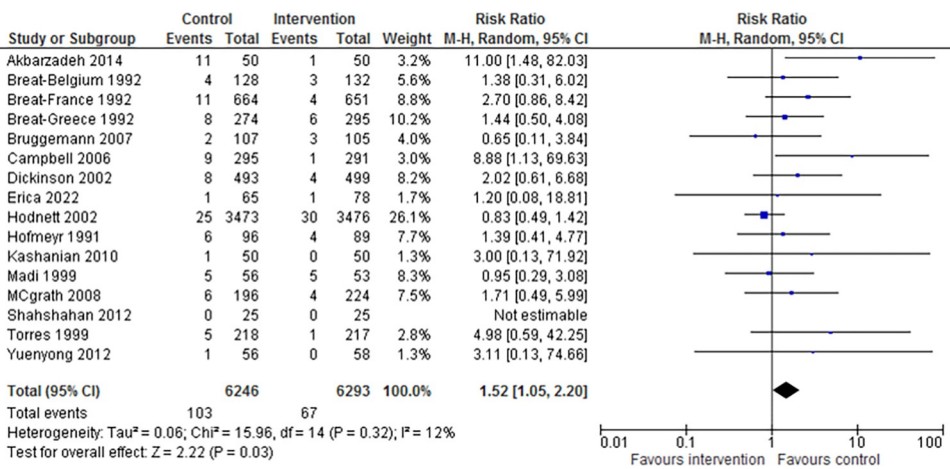

**Fig 4. Forest plot for the meta-analysis of 5-minute Apgar score $< 7$.**

European region. However, there was an uneven distribution of trials and participants between subgroups in all analyses, with the highest number of participants in the European subgroup.

## Discussion

### Research gap

The advantages of continuous labor companionship are well recognized. However, implementing a labor companion policy lags behind the evidence and is met with significant resistance at all levels [45]. There is a wide range of definitions, outcomes, and indicators used to describe support by a labor companion, and there is a reporting heterogeneity leading to divided opinions. Also, scarce evidence is available to define the ideal labor companion.

Confounding factors affecting the total labor experience sometimes undermine the importance of having a labor companion. Therefore, different levels of labor companions, from family members to trained healthcare workers, continue to play a role in obstetrics care delivery systems worldwide. This heterogeneity, while coloring the opinion within the local setting, seems to affect the quality of research published and thus leads to a lack of higher-level evidence on which healthcare managers can rely to support the use of "the best" labor companion. Our study attempts to standardize the optimum characteristics of the ideal labor companion, potentially leading to better labor experience across all settings, lesser labor complication rates, better cost-effectiveness, and, hopefully, universal implementation.

### Summary of main results

This review included 35 RCTs involving 16414 participants, conducted in hospital settings in 18 countries, highlighting significant positive effects of continuous labor support across various outcomes. The primary analysis showed the highest overall effect in the improvement reported in the 5-minute Apgar score $< 7$, with an effect size of 1.52 (95% CI 1.05, 2.20). Also encouraging is that the mere presence of a labor companion improved all eight studied outcomes compared to the absence, regardless of subgrouping characteristics. The secondary analysis showed that familiar labor companions were better at reducing tocophobia, with an effect size of 1.73 (95% CI 1.49, 2.42), compared to unfamiliar companions, with an effect size of 1.34 (95% CI 1.14, 1.58). Untrained labor companions were the better choice in reducing tocophobia and the cesarean section rate compared to trained companions. For the analysis of

**Table 3. Effectiveness of having a familiar versus unfamiliar labor companion.**

| Outcome | | No. of Participants (Studies) | RR (95% CI) | P value | Heterogeneity (I²) | Test for subgroup difference (p-value) |
|---|---|---|---|---|---|---|
| 1. Spontaneous vaginal delivery | Familiar | 1706 (7 RCTs) | 1.12(0.99,1.28) | 0.07 | 80% | 0.6 |
| | Unfamiliar | 12105 (12 RCTs) | 1.08(1.04,1.13) | 0.0002 | 60% | |
| 2. Duration of labor (Hours) | Familiar | 1335 (4 RCTs) | 0.31(0.20,0.41) | 0.0001 | 0% | 0.94 |
| | Unfamiliar | 4087 (13 RCTs) | 0.31(0.16,0.46) | 0.0001 | 77% | |
| 3. Cesarean section | Familiar | 2284 (8 RCTs) | 1.55(1.00,2.39) | 0.05 | 76% | 0.63 |
| | Unfamiliar | 12796 (16 RCTs) | 1.38(1.14,1.67) | 0.001 | 54% | |
| 4. Instrumental delivery | Familiar | 1752 (8 RCTs) | 1.12(0.95,1.32) | 0.19 | 22% | 0.81 |
| | Unfamiliar | 12203 (13 RCTs) | 1.15(1.01,1.30) | 0.03 | 31% | |
| 5. Oxytocin for labor induction | Familiar | 2289 (8 RCTs) | 1.09(0.93,1.27) | 0.28 | 79% | 0.39 |
| | Unfamiliar | 10669 (13 RCTs) | 1.20(1.03,1.38) | 0.02 | 78% | |
| 6. Analgesic usage | Familiar | 2075 (7 RCTs) | 1.07(0.97,1.18) | 0.21 | 61% | 0.95 |
| | Unfamiliar | 10644 (11 RCTs) | 1.07(1.01,1.14) | 0.03 | 48% | |
| 7. Tocophobia | Familiar | 1708 (4 RCTs) | 1.73(1.49,2.02) | 0.0001 | 29% | 0.02 |
| | Unfamiliar | 9425 (7 RCTs) | 1.34(1.14,1.58) | 0.0004 | 48% | |
| 8. 5 min Apgar < 7 | Familiar | 1749 (8 RCTs) | 2.40(0.93,6.17) | 0.07 | 36% | 0.21 |
| | Unfamiliar | 10790 (8 RCTs) | 1.25(0.88,1.79) | 0.22 | 0% | |

RCT–Randomized Controlled Trial, RR–Relative Risk, CI–Confidence Interval

tocophobia, the pooled effect sizes were 1.34 (95% CI 1.14, 1.57) and 1.84 (95% CI 1.60, 2.12) in trained versus untrained subgroup comparisons. For the cesarean rate, they were represented as 1.22 (95% CI 1.05, 1.42) and 2.16 (95% CI 1.37, 3.40), respectively. The pooled effect size for the duration of labor was 0.16 (95% CI 0.06, 0.26) for the subgroup of RCTs conducted before 2000 and 0.53 (95% CI 0.30, 0.77) for the subgroup of RCTs conducted after 2000. A significant subgroup difference (<0.1) was found in relation to the duration of labor, cesarean section rate, oxytocin for labor induction, analgesic usage, and tocophobia in the subgroup analysis of geographical regions, indicating that 'geographical setting' modifies the outcomes in a statistically significant way.

## Impact of labor companionship on emergency cesarean section rates

Cesarean section rates continued to rise through the last few decades, resulting in significant medicalizing of labor and higher healthcare costs with doubtful accruing benefits [46]. Although labor companionship has a low to moderate effect on reducing cesarean sections [10, 17, 29, 31], it is a useful strategy to incorporate into a broader plan. By overlapping

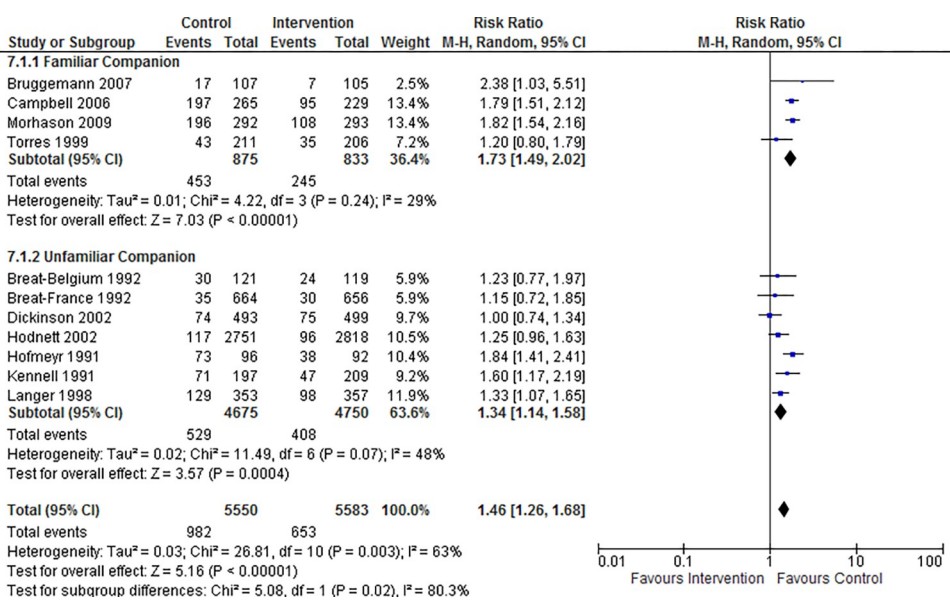

**Fig 5. Forest plot for the subgroup analysis of tocophobia.**

beneficial effects of lowering tocophobia, the ability of effectively directed pushing, and shorter labor duration, the contribution of a labor companion towards reducing cesarean section rates is incontrovertible [3, 7, 32, 42].

Tocophobia increases the stress and sympathetic response in the mother during labor. Maternal stress reactions are associated with fetal tachycardia and variable decelerations [5, 6]. The resultant cardiotocography changes may lead to high cesarean section rates. As the management protocols used during abnormal CTG events during the studies were unavailable, the effect could not be analyzed further in the available dataset.

## Impact of labor companionship on neonatal outcome

Apgar score at 5 minutes is more explicit of intrapartum fetal condition [47], which is improved by having a labor companion; therefore, labor companionship can help reduce neonatal intensive care unit and special care baby unit admissions. This is likely a cumulative effect of less stressful labor with lower maternal catecholamine levels. Our results revealed a significant improvement in the five-minute Apgar score in the presence of a labor companion, irrespective of the type of companion. Several studies have looked at the possibility of continuing the support provided by the labor companion into neonatal care support and beyond [9, 16, 36]. The authors agree that the role of the labor companion does not have to stop at the baby's delivery; it may continue through the first few months of motherhood and recommend further directed research into the topic.

## Impact of labor companionship on intrapartum and postpartum psychological well-being

The presence of a labor companion during childbirth has been shown to improve maternal satisfaction [17, 24, 26, 28], boost maternal self-esteem [26, 38], reduce postpartum depression [38, 40, 42], reduce anxiety [7, 13, 32] and improve overall psychological well-being [6, 21, 36]. Due to the heterogeneity of tools used for assessing maternal satisfaction, self-esteem, and postpartum depression, meta-analysis was not possible. However, tocophobia was reported

**Table 4. Effects of having a labor companion in different geographical regions.**

| Outcome | | No. of Participants (Studies) | RR (95% CI) | P value | Heterogeneity (I²) | Test for subgroup difference (p-value) |
|---|---|---|---|---|---|---|
| 1. Spontaneous vaginal delivery | Asia | 464 (4 RCTs) | 1.18(0.84,1.64) | 0.35 | 94% | 0.77 |
| | Africa | 298 (2 RCTs) | 1.13(0.92,1.39) | 0.24 | 67% | |
| | Europe | 2366 (5 RCTs) | 1.06(1.01,1.11) | 0.02 | 0% | |
| 2. Duration of labor | Asia | 327 (4 RCTs) | 0.67(0.16,1.19) | 0.002 | 80% | 0.02 |
| | Africa | 774 (2 RCTs) | 0.28(0.07,0.48) | 0.008 | 41% | |
| | Europe | 2374 (5 RCTs) | 0.09(-0.03,0.21) | 0.15 | 40% | |
| 3. Cesarean section | Asia | 386 (4 RCTs) | 2.57(1.29,5.12) | 0.007 | 44% | 0.02 |
| | Africa | 883 (3 RCTs) | 2.02(1.16,3.52) | 0.01 | 42% | |
| | Europe | 2509 (6 RCTs) | 1.12(0.84,1.48) | 0.44 | 0% | |
| 4. Instrumental delivery | Asia | 264 (3 RCTs) | 0.86(0.47,1.56) | 0.62 | 0% | 0.43 |
| | Africa | 298 (2 RCTs) | 1.82(0.42,7.93) | 0.43 | 64% | |
| | Europe | 2509 (6 RCTs) | 1.21(1.05,1.39) | 0.009 | 0% | |
| 5. Oxytocin for labor induction | Asia | 427 (5 RCTs) | 1.16(0.95,1.43) | 0.15 | 0% | 0.07 |
| | Africa | 883 (3 RCTs) | 1.24(0.83,1.84) | 0.29 | 35% | |
| | Europe | 2373 (5 RCTs) | 0.99(0.88,1.10) | 0.81 | 40% | |
| 6. Analgesic usage | Asia | 114 (1 RCT) | 1.20(0.63,2.28) | 0.59 | NA | 0.04 |
| | Africa | 883 (3 RCTs) | 1.13(0.94,1.34) | 0.18 | 25% | |
| | Europe | 1965 (5 RCTs) | 1.10(1.01,1.21) | 0.04 | 0% | |
| 7. Tocophobia | Asia | NA | NA | NA | NA | 0.003 |
| | Africa | 773 (2 RCTs) | 1.83(1.58,2.11) | 0.0001 | 0% | |
| | Europe | 1560 (2 RCTs) | 1.19(0.85,1.67) | 0.31 | 0% | |
| 8. 5 min Apgar < 7 | Asia | 364 (4 RCTs) | 6.22(1.39,27.84) | 0.02 | 0% | 0.58 |
| | Africa | 294 (2 RCTs) | 1.14(0.49,2.67) | 0.77 | 0% | |
| | Europe | 2287 (4 RCTs) | 1.74(0.90,3.38) | 0.10 | 0% | |

RCT–Randomized Controlled Trial, RR–Relative Risk, CI–Confidence Interval

homogeneously among different studies. Tocophobia is a form of extreme anxiety about labor that, in some, may amount to post-traumatic stress [2]. The current study showed that familiar and untrained labor companions better alleviate tocophobia than unfamiliar and trained companions. A familiar labor companion/companion of choice may have a close emotional bond with a laboring mother compared to a newly introduced unfamiliar labor companion. Formal training as a labor companion may attribute distance in the emotional bond when adhering to the instructed protocols, increasing anxiety. Studies have shown that labor companionship can affect the long-term psychological well-being of the mother [38, 40, 42], which is a potential area for further research.

## Comparative effectiveness of familiar versus unfamiliar labor companionship

Comparing familiar and unfamiliar labor companions revealed a larger effect size in our study, favoring familiar labor companions in achieving successful vaginal delivery, having shorter labor, lower section rates, lower tocophobia rates, and fewer neonates with Apgar <7 at 5 minutes. Familiarity and attachment between the laboring mother and the companion are likely to provide better support and meet the expectations of the parturient during labor. It is uncommon to have a familiar labor companion who is also trained. Therefore, it is more likely that the effect is due to familiarity rather than the training they have received. Some studies observed that with familiar labor companions, women are less likely to adopt optimal positions during labor which are proven to shorten labor duration and reduce assisted vaginal deliveries [3]. They are more likely to stay in the most comfortable position, which sometimes may be counterproductive. With trained support, women tend to stay in lateral or upright positions during labor without epidurals and lateral positions with epidurals [4]. However, our results suggest that this criticism has no significant impact and is no reason to deny a companion of choice.

While most studies use low-risk mothers to mitigate the difficulties in exploring labor with medical and fetal complications, according to the authors' best knowledge, there are no RCTs involving mothers with complicated pregnancies. Pregnant mothers with heart diseases would likely find it beneficial in terms of cardiac status to have less pain and less anxiety with a familiar labor companion. We feel the impact of labor companions in medically complicated pregnancies is an area needing urgent further exploration.

## Training of labor companions versus making trained labor companions familiar

Current consensus suggests extending midwifery care to ensure the likelihood of having a familiar carer during labor. From early pregnancy, closer, more personal, coordinated shared care by the community and hospital-based midwives have the potential to make a familiar, trained labour caregiver available during the labor.

However, this does not replace the role of a labor companion. Labour companion, by definition, is not involved in the medical needs of the parturient. A fully trained and qualified labour caregiver is a luxury even in developed economies, and the shortage of midwives is acute globally. A labour caregiver as a companion is unlikely to be economically viable.

A viable strategy would be to select a familiar labor companion early in the pregnancy and provide suitable training to them integrated into the existing antenatal education program conducted by hospital-based and field midwives.

It is unclear whether a trained labour companion provides more effective cover than simply a familiar labour companion or whether the women should choose a familiar companion and

that companion can undergo training for the role. Developing such training programs may not be cost-effective, considering the significant advantages of having an untrained, familiar labor companion per laboring mother with trained staff to oversee the entire process. It is clear from other studies comparing psychological and emotional responses toward laboring women that training should include a psychological component and highlight empathy as one of the most important characteristics of a labor companion, and such training is hard to implement [7, 13, 32, 38, 40]. Further, there should be regular validation, certification, and continuous professional development for these trained labor companions.

There is insufficient data to assess the similarities of labor companion training associated with the included studies. Only some papers gave short descriptions of how the training was conducted [34, 35, 37, 39]. Also, healthcare workers with different levels of training and experience were employed as labor companions in the included studies [5, 25, 28, 29]. Determining minimum labor companion training standards within the current analysis is challenging.

In summary, it is unclear whether the training for the role affects a labour companion's effectiveness. Also unclear is the relative importance of familiarity versus the training labour companions receive. Familiar companions tend not to have received any training, but they seem effective based on an indirect analysis of the results. Whether the training being provided somehow reduces the effectiveness of labour companions who would otherwise be more effective is also a consideration. Overall, this area needs more research to understand the relationship.

## Choosing the ideal support person

There is no evidence to determine who would be the best labor companion. The available studies involved friends, female relatives, healthcare professionals, or community personnel. Only one study used a partner as a labor companion [7], which may be due to cultural reasons or privacy issues of other women in the labor care setting or lack of invitation for male partners to participate in the labor process actively. This is a potential area for further research. It would be interesting to compare different relatives acting as labor companions. Due to complex relationship dynamics and cultural differences, it would be very difficult to conduct and interpret such a study. Some cultures are more reluctant to accept the male partner as the labor companion. In resource-poor labor room settings, it would be unethical to do so if it is difficult to maintain the privacy of laboring women. High heterogeneity is a limitation of this study in providing recommendations on the ideal labor companion. Presently, local protocols accounting for local cultural beliefs would be of superior benefit.

The current resource-poor labor room settings do not give any room for direct family involvement. Due to limitations in funding and staffing, there are hardly any waiting areas for the family to stay during the labor of their loved one. The labor companion may also require support during their role, and waiting areas may just be what they need to wind down. On the flip side, we also need to consider the ramifications of such an arrangement in the spectrum of worldwide social norms.

## Temporal evolution of study data

The temporal differences between studies conducted before 2000 and later, as demonstrated in our research, can be due to the recorded rise of cesarean section rates after 2000. Also, with international consensus, there has been a marked improvement in the methodological quality of randomized controlled trials since 2000, especially with the introduction of trial registries in 2006. The findings may also be due to changes in labor management guidelines and better availability of fetal and maternal monitoring facilities. More spontaneous vaginal deliveries

may have occurred before the year 2000, while with a modern understanding of fetal physiology and the availability of continuous electronic monitoring, interventions may have become more likely. With the previous linear understanding of labor durations, more cesarean sections may have occurred due to a suspected lack of progress compared to the current dynamic approach to stages of labor.

## Comparative effectiveness according to geographic location

Differences in effect sizes of secondary outcomes between Asia, Africa, and Europe may be observed due to regional differences in obstetric practice. Asia and Africa, accounting for most of the world's lower-middle-income economies, lack some basic facilities widely available on the European continent. Obstetricians, therefore, make decisions based on the overall situation rather than following strict guidelines. In using the positive influence of labor companionship, it is not surprising that more cesarean deliveries can be prevented in Asian and African regions compared to Europe. Since the cost of a cesarean delivery is much higher compared to that of a vaginal delivery, the opportunity-cost saving would be much higher [48].

There are no studies assessing the acceptability of having a labor companion. There are established acceptability constructs that can address both patients' and healthcare workers' points of view [49]. Well-conducted studies would provide further evidence for incorporating the practice into routine labor care everywhere, especially in low-resource settings. Also, attitudes towards having a labor companion must be assessed based on established questionnaires. While no such studies exist, further research would be of enormous value.

Creating awareness among the public would be essential in incorporating this paradigm-shifting practice into routine obstetric care. It would circumvent any possible backlash from different mindsets and social backgrounds.

## Strengths and limitations

We harnessed the strength of rigorous research, utilizing data from 35 RCTs, and delved deeply into the nuances of labor companionship, scrutinizing factors like familiarity versus unfamiliarity, trained versus untrained companions, and temporal associations—an approach not previously explored in existing meta-analyses [50]. This comprehensive examination provides valuable insights into the impacts of labor companionship that go beyond what has been previously studied.

Labor companions have been proven to significantly benefit fetal and maternal health in various setups and settings. However, universal uptake is lagging. The causes for the lagging can be multifactorial, where problems with implementation are partly responsible [45]. Some research questions concerning the role of labor companions demand high-quality evidence, typically sourced from RCTs. Conversely, questions regarding maternal satisfaction and staff attitudes toward labor companions usually do not warrant RCTs or can be adequately addressed using alternative methodologies. We conducted a thorough review of RCTs to provide answers where rigorous evidence was crucial while also employing meta-analysis to bridge research gaps.

The current study does come with its limitations. All 35 RCTs included in our analysis exhibited a high degree of bias due to unblinding, as blinding is impossible due to the nature of the intervention, a necessary limitation. We used the older original version of the Cochrane Risk of Bias tool rather than the more recent version 2 of the Cochrane Risk of Bias tool (RoB-02). Due to significant heterogeneity in reporting across the trials, we could only identify eight distinct outcomes with sufficient studies, limiting our ability to perform robust subgroup analyses.

All the analyzed RCTs, especially the three with the highest weight, had spontaneous onset of labor as an inclusion criterion. Therefore, the results may not be applicable in an induced labor setting. Since spontaneous labor leads to more successful vaginal deliveries, less use of oxytocin augmentation, less assisted vaginal deliveries, shorter labors, and less analgesia use compared to induced labor, the effect of a labor companion alone may not be so profound in this analysis. Studies comparing similar outcomes between spontaneous and induced labor in the presence of a labor companion would help clarify the issue.

## Recommendations

Future research into labor companionship should prioritize conducting well-designed RCTs with a rigorous methodology to mitigate bias and improve the quality of evidence in this area. Additionally, efforts should be made to standardize reporting practices to enhance comparability across studies and facilitate more extensive subgroup analyses. This would result in minimal heterogeneity, paving the way to a robust meta-analytical evidence base to form recommendations. In the interim, healthcare providers should engage in shared decision-making with expectant mothers, considering their preferences and individual circumstances when considering labor companions.

Well-designed qualitative studies that assessed satisfaction with labor, acceptability, and stress experienced during labor using validated tools and quantification methods like the Likert Scale would add significant value to the evaluation of the psychological and emotional component of the labor process.

## Conclusion

Labor companionship demonstrates potential benefits across various maternal and neonatal outcomes. However, the type of best labor companion remains inconclusive. While a labor companion may not have as profound an effect in spontaneous labor compared to induced labor scenarios, it still contributes to improved birthing experiences and reduced cesarean section rates. A familiar, untrained companion offers greater support during labor, emphasizing the importance of emotional connection. However, there are limitations, including study heterogeneity, a lack of data on companion training, and temporal differences in study outcomes. Despite these limitations, labor companionship can be a valuable strategy to incorporate into broader obstetric care plans.

## Supporting information

**S1 Table. Effectiveness of trained versus untrained labor companion.**
(DOCX)

**S2 Table. Effectiveness of a labor companion before and after 2000.**
(DOCX)

**S1 File. Analysis of spontaneous vaginal delivery.**
(DOCX)

**S2 File. Analysis of duration of labor.**
(DOCX)

**S3 File. Analysis of cesarean section.**
(DOCX)

**S4 File. Analysis of instrumental delivery.**
(DOCX)

**S5 File. Analysis of oxytocin for labor induction.**
(DOCX)

**S6 File. Analysis of analgesic usage.**
(DOCX)

**S7 File. Analysis of tocophobia.**
(DOCX)

**S8 File. Analysis of 5 min Apgart $<$ 7.**
(DOCX)

**S9 File. PRISMA 2020 checklist.**
(DOCX)

**S10 File. Detailed search strategy.**
(DOCX)

**S11 File. Study protocol.**
(PDF)

**S12 File. Summary data.**
(DOCX)

## Author Contributions

**Conceptualization:** D. M. C. S. Jayasundara, I. A. Jayawardane, S. D. S. Weliange, T. D. K. M. Jayasingha, T. M. S. S. B. Madugalle.

**Data curation:** D. M. C. S. Jayasundara, I. A. Jayawardane, S. D. S. Weliange, T. D. K. M. Jayasingha, T. M. S. S. B. Madugalle.

**Formal analysis:** D. M. C. S. Jayasundara, S. D. S. Weliange, T. D. K. M. Jayasingha.

**Funding acquisition:** D. M. C. S. Jayasundara.

**Investigation:** D. M. C. S. Jayasundara, I. A. Jayawardane.

**Methodology:** D. M. C. S. Jayasundara, I. A. Jayawardane, S. D. S. Weliange, T. D. K. M. Jayasingha, T. M. S. S. B. Madugalle.

**Project administration:** D. M. C. S. Jayasundara.

**Resources:** D. M. C. S. Jayasundara.

**Software:** D. M. C. S. Jayasundara, I. A. Jayawardane, T. D. K. M. Jayasingha.

**Supervision:** D. M. C. S. Jayasundara, I. A. Jayawardane, S. D. S. Weliange.

**Validation:** D. M. C. S. Jayasundara, I. A. Jayawardane, S. D. S. Weliange, T. M. S. S. B. Madugalle.

**Visualization:** D. M. C. S. Jayasundara, I. A. Jayawardane.

**Writing – original draft:** D. M. C. S. Jayasundara, T. D. K. M. Jayasingha, T. M. S. S. B. Madugalle.

**Writing – review & editing:** D. M. C. S. Jayasundara, I. A. Jayawardane, S. D. S. Weliange, T. D. K. M. Jayasingha.

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
