## [Decision Letter · Decision Letter 0]

26 Mar 2024

PONE-D-24-02940Impact of continuous labour companion- who is the best: A comprehensive meta-analysis on familiarity, training, temporal association, and geographical locationPLOS ONE

Dear Dr. Jayasundara,

Thank you for submitting your manuscript to PLOS ONE. After careful consideration, we feel that it has merit but does not fully meet PLOS ONE’s publication criteria as it currently stands. Therefore, we invite you to submit a revised version of the manuscript that addresses the points raised during the review process.

**The reviewers have suggested some revisions to your article. Please address these according to the instructions below.**

We look forward to receiving your revised manuscript.

Kind regards,

Tanya Doherty, PhD

Academic Editor

PLOS ONE

Reviewers' comments:

Reviewer's Responses to Questions

**Comments to the Author**

1. Is the manuscript technically sound, and do the data support the conclusions?

Reviewer #1: Yes

Reviewer #2: Yes

2. Has the statistical analysis been performed appropriately and rigorously? 

Reviewer #1: I Don't Know

Reviewer #2: Yes

3. Have the authors made all data underlying the findings in their manuscript fully available?

Reviewer #1: Yes

Reviewer #2: Yes

4. Is the manuscript presented in an intelligible fashion and written in standard English?

Reviewer #1: Yes

Reviewer #2: Yes

5. Review Comments to the Author

Reviewer #1: Thank you for the opportunity to review this article. I enjoyed reading about this interesting study. The article described the results of a meta-analysis of 35 RCTs on labour companionship and maternal and neonatal outcomes. The study aimed to determine if labour companion training or familiarity had an effect on these outcomes. Overall, the article is well-written and presented and I felt that it would make a useful contribution to the literature. My main concern was with the discussion section which I felt should be revised to include a more substantial engagement with the existing literature in this area.

Introduction

The introduction provides a useful overview of the emotional labour experience and how fear and anxiety may play a role in labour outcomes as well as the role of labour companions in improving the labour experience. A clear rationale for the current study was presented. My only suggestions here would be to include definitions for some of the key concepts which are used throughout the article. Eg tokophobia and to perhaps reconsider the use of the term ‘developing countries’ possibly in favour of LMICs?

Method

Generally, I felt that the necessary information was reported in this section.

Results

The results of the analysis were clearly presented with helpful tables complementing the text. I do not have the expertise to assess the rigour of the statistical analysis and so would recommend an additional reviewer with the requisite experience evaluate this.

Discussion

The discussion was interesting, however, my main concern is the lack of literature in this section. I also had a few more minor questions and suggestions.

Tokophobia vs tokophobia- one spelling should be used consistently.

The possible mechanisms through which labour companionship may be influencing APGAR score could be included on p19 lines 310-311.

References to the studies being referred to on line 316 should be included on p19.

Some of the definitions and explanations in the discussion section could be included in the introductory section for example lines 317-320 on p19.

Lines 332-336. The discussion of the role of familiar vs unfamiliar labour companions on birthing positions was interesting but again references to the other studies referred to should be included- (lines 332-336) and I felt that the links to the data in the present study could be made clearer. The implications for future research/practice of this observation could also be unpacked.

Lines 343 and 347, the authors mention ‘the current resource poor labour room setting’ it may be helpful to mention that you are referring to the many contexts where resources are limited as at the moment it implies that everywhere is resource poor.

The discussion section as a whole could benefit from a deeper engagement with the literature, I don’t think there are any references in this section. Many of the claims made here are not supported with references to existing studies in this area which would significantly strengthen the arguments being made.

Recommendations

Perhaps the authors could acknowledge the challenges of reducing bias in studies like the ones included in this article. I also wondered about how qualitative studies exploring the perspectives of birthing women and labour companions might compliment the RCTs on this topic.

Conclusion

The conclusion provided a good summary of some of the key points made in the article although the recommendation to standardize companion training did not feature in the recommendations section and was not really indicated from the results of the study and so could perhaps be removed.

Reviewer #2: Manuscript #: PONE-D-24-02940

Title: Impact of continuous labour companion- who is the best: A comprehensive meta-analysis on familiarity, training, temporal association, and geographical location

Authors: DMCS Jayasundara; IA Jayawardane; SDS Weliange; TDKM Jayasingha; TMSSB Madugalle

Article type: Research Article

Thank you for the opportunity to review this important work.

I should congratulate the authors on this meticulous work, which is publishable in PLOS ONE. However, I would like to suggest a few comments to improve the scientific validity and readability of the manuscript.

Title

Suggest adding a ‘systematic review and meta-analysis of randomised controlled trials’ to the title. Please make sure the title is not too long.

Abstract

Mention the timeline of your search.

Please explain how funnel plots are used to assess the risk of bias. Funnel plots are used to assess the publication bias.

Prospective registration of the protocol indicates the transparency of the work. Could you please mention whether this work was registered prospectively? If not, explain.

CI – You need to word it out before using it as an abbreviation.

APGAR – This is not an abbreviation. Regina Apgar introduced this score. Please correct this as Apgar throughout the manuscript.

Line 40-43: Please add the summary statistics for the results. There is a difference between a lay language summary and a scientific abstract. Please have a look at a Cochrane review of the way they have presented the results.

Line 44: Add systematic review and meta-analysis. It is unclear why the authors mentioned underscoring the benefits of labour companionship. It is not clear; please rephrase.

The discussion/conclusion needs to be rephrased for clarity.

Main text

Line 96-97: Tokophobia and tokophobia are used interchangeably. Please use one. Please provide a definition for tocophobia.

As this topic is interesting and broad, it would be great if the authors could introduce a few subheadings to each section. The introduction can be divided into background/literature review/gap of knowledge/objectives, etc. The same applies to the other sections: methods, results, and discussion.

Line 106: was the search from inception to 04-07-2023? Please mention.

Line 177: Can you provide references for these meta-analyses? Perhaps you could discuss these with your findings in the discussion section.

Line 118: I suggest this part should go into a subheading, such as screening eligible studies. Please mention the initials of each author responsible for reviewing. Generally, conflicts between the primary and second reviewers are resolved by a third reviewer, if required.

Please add a sub-title explaining all the categories: familiar versus unfamiliar, trained versus untrained, and what temporal association is.

How did you divide the geographical territory? WHO has a categorization.

Line 130: add ‘viable’ singleton pregnancies

Line 130: early labour, more accurately, the latent phase.

Line 132: please explain what you mean by genital abnormalities here

Lines 133-134: What about quasi-experimental studies? Otherwise, you can simply mention that any other study designs, other than randomised controlled trials, including quasi-experimental trials, were excluded. Did you have limitations for the article language? Were all the articles in English?

Line 135: Suggest not mixing the study outcomes with the data extraction and quality assessment. Study outcomes should be a separate subtopic followed by another subtopic for the ‘Risk of bias and quality assessment’- RoB2 and possible biases. In the study outcomes, please mention primary and secondary outcomes clearly.

Line 144-146: Risk of bias assessment – I am not sure the authors have used the current RoB -2 tool as the domains that appear on the manuscript are different from the RoB -2. Looks like authors have used RoB -01. Please update accordingly.

Please go to this link for more information on RoB 2: https://sites.google.com/site/riskofbiastool/welcome/rob-2-0-tool/current-version-of-rob-2

Line 150-160: Suggest using ‘versus’ instead of ‘vs’.

Table 1: Please provide what LSCS means as a footnote. All tables/figures should be self-explanatory. Can not use RCT in the table legend. Please word it out.

Table 1: What is the order of appearance of the RCTs in Table 1? It should be either the chronological order of appearance, A-Z or any other rational categorization.

Table 1, 6th column: what are these outcomes? Are these the primary outcomes reported by the trials? Or else all the outcomes? It would be more explanatory to divide primary and secondary outcomes.

Line 219: There cannot be more than one funnel plot to assess the publication bias for one meta-analysis. Could you please elaborate more?

Line 220-221: Inability to blind is a limitation; please acknowledge this in the discussion.

In Tables 2/3/4, I2 for heterogeneity appears as a decimal value. Is this a mistake? According to the Cochrane Handbook, I2 should be 0-100% as a percentage.

Line 260: cesarean section, which is abbreviated as LSCS. Please correct.

Suggest American English terms like labor, not ‘labour’ throughout the manuscript.

Table 4: 6th column – please explain what you mean by the test for subgroup difference.

Discussion:

- The first paragraph needs to be rewritten for clarity. It should start with the research gap and objective and end with the main findings.

- Please divide the entire discussion into separate sections as sub-topics to increase the readability of the paper. Please follow the steps of a well-written systematic review publication in a reputed journal. For example: Main findings, Critical review of literature, Strengths and limitations, Conclusions, Recommendations, Future directions etc.

- Line 312: Define ICU/SCBU abbreviations

Reference list:

None of the references have included the journal name.

I hope this might improve this work towards publication.

Thank you.

6. PLOS authors have the option to publish the peer review history of their article (what does this mean?). If published, this will include your full peer review and any attached files.

Reviewer #1: No

Reviewer #2: No

---

## [Author Response · Author response to Decision Letter 0]

9 May 2024

We thank the reviewers and the editorial office for the excellent comments, suggestions, and advice on improving our meta-analysis. We have accepted all comments, and the article was amended accordingly. Details of the amendments are given below. 

Comments to the Author

Reviewer #1: Thank you for the opportunity to review this article. I enjoyed reading about this interesting study. The article described the results of a meta-analysis of 35 RCTs on labour companionship and maternal and neonatal outcomes. The study aimed to determine if labour companion training or familiarity had an effect on these outcomes. Overall, the article is well-written and presented and I felt that it would make a useful contribution to the literature.

My main concern was with the discussion section which I felt should be revised to include a more substantial engagement with the existing literature in this area.

We thank the reviewer for the encouraging comments. We have revised the discussion, including a substantial engagement with the literature. We have rephrased the suggested areas for clarity. We have added definitions and possible mechanisms as instructed (Highlighted). 

Introduction

The introduction provides a useful overview of the emotional labour experience and how fear and anxiety may play a role in labour outcomes as well as the role of labour companions in improving the labour experience. A clear rationale for the current study was presented. 

My only suggestions here would be to include definitions for some of the key concepts which are used throughout the article. Eg tokophobia and to perhaps reconsider the use of the term 'developing countries' possibly in favour of LMICs?

Response: We thank the reviewer for the comments. We have added a definition for tocophobia (Lines 65-66), labor companion (Lines 84-86), and doula (Line 84). We have also changed the term developing countries to LMIC (Line 87).

Method

Generally, I felt that the necessary information was reported in this section.

Results

The results of the analysis were clearly presented with helpful tables complementing the text. I do not have the expertise to assess the rigor of the statistical analysis and so would recommend an additional reviewer with the requisite experience evaluate this.

Discussion

The discussion was interesting, however, my main concern is the lack of literature in this section. I also had a few more minor questions and suggestions.

Response – Thank you. Citations to studies mentioned in the discussion segment have been added, and the discussion expanded to reflect engagement with the current literature. 

Tokophobia vs tocophobia- one spelling should be used consistently.

Response – Thank you. Tocophobia has been used as the spelling of choice.

The possible mechanisms through which labour companionship may be influencing APGAR score could be included on p19 lines 310-311.

Response – Thank you. We have amended the discussion with the possible mechanisms mentioned (Line 377-378).

References to the studies being referred to on line 316 should be included on p19.

Response – Thank you. Included within the citations/bibliography (Lines 386-388)

Some of the definitions and explanations in the discussion section could be included in the introductory section for example lines 317-320 on p19.

Response – Thank you. The definition of tocophobia has been included in the introduction (Lines 65-66)

Lines 332-336. The discussion of the role of familiar vs unfamiliar labour companions on birthing positions was interesting but again references to the other studies referred to should be included- (lines 332-336) and I felt that the links to the data in the present study could be made clearer. 

Response – Thank you. We have rephrased for clarity, and references were added (Line 403-409)

The implications for future research/practice of this observation could also be unpacked.

Response – Thank you. We have discussed implications for future research/practice in a separate paragraph. (Line 409-412) 

Lines 343 and 347, the authors mention 'the current resource poor labour room setting' it may be helpful to mention that you are referring to the many contexts where resources are limited as at the moment it implies that everywhere is resource poor.

 Response – Thank you. We agree. Rephrased for clarity (Line 451 and 456)

The discussion section as a whole could benefit from a deeper engagement with the literature, I don't think there are any references in this section. Many of the claims made here are not supported with references to existing studies in this area which would significantly strengthen the arguments being made.

Response – Thank you. We have supported the claims with existing evidence (Citations added)

Recommendations

Perhaps the authors could acknowledge the challenges of reducing bias in studies like the ones included in this article. I also wondered about how qualitative studies exploring the perspectives of birthing women and labour companions might compliment the RCTs on this topic.

Response – We have reviewed the recommendations and added a discussion on possible value addition by qualitative studies (Lines 521 – 524)

Conclusion

The conclusion provided a good summary of some of the key points made in the article although the recommendation to standardize companion training did not feature in the recommendations section and was not really indicated from the results of the study and so could perhaps be removed.

Response – We agree. Removed. 

Reviewer #2: Manuscript #: PONE-D-24-02940

Title: Impact of continuous labour companion- who is the best: A comprehensive meta-analysis on familiarity, training, temporal association, and geographical location

Authors: DMCS Jayasundara; IA Jayawardane; SDS Weliange; TDKM Jayasingha; TMSSB Madugalle

Article type: Research article

Thank you for the opportunity to review this important work. I should congratulate the authors on this meticulous work, which is publishable in PLOS ONE. However, I would like to suggest a few comments to improve the scientific validity and readability of the manuscript.

Title

Suggest adding a 'systematic review and meta-analysis of randomised controlled trials' to the title. Please make sure the title is not too long.

Response: We thank the reviewer for the encouraging evaluation and the suggestions for improvement. We have changed the title to, 

Title: The impact of continuous labor companion- who is best: A systematic review and meta-analysis of randomized controlled trials.

Abstract

Mention the timeline of your search.

Response: We have added the timeline of our search (Line 29)

Please explain how funnel plots are used to assess the risk of bias. Funnel plots are used to assess the publication bias.

Response: We acknowledge that the funnel plots are used to assess the risk of publication bias and not to assess the risk of bias. We have amended accordingly (Line 31)

Prospective registration of the protocol indicates the transparency of the work. Could you please mention whether this work was registered prospectively? If not, explain.

Response: Thank you. We agree that the registration of systematic reviews has multiple scientific merits and includes prospective registration, strengthening transparency. The current study was rigorous in methodology and registered as an ongoing project with INPLASY. (Registration number: INPLASY202410003).

CI – You need to word it out before using it as an abbreviation.

Response: Thank you. We have corrected as instructed (Line 39)

APGAR – This is not an abbreviation. Regina Apgar introduced this score. Please correct this as Apgar throughout the manuscript.

Response: Thank you. We have corrected it as instructed.

Line 40-43: Please add the summary statistics for the results. There is a difference between a lay language summary and a scientific abstract. Please have a look at a Cochrane review of the way they have presented the results.

Response: Thank you. We have added summary statistics (Line 41-50)

Line 44: Add systematic review and meta-analysis. Amended discussion (Lines 51-58) 

It is unclear why the authors mentioned underscoring the benefits of labour companionship. It is not clear; please rephrase. The discussion/conclusion needs to be rephrased for clarity.

Response: We acknowledge the reviewer's comment. We have amended it for a clearer discussion/conclusion (Lines 51-58) 

Main text

Line 96-97: Tokophobia and tocophobia are used interchangeably. Please use one. Please provide a definition for tocophobia.

Response: We have changed to 'tocophobia' throughout the manuscript. Tocophobia is defined in the introduction (Lines 65-66)

As this topic is interesting and broad, it would be great if the authors could introduce a few subheadings to each section. The introduction can be divided into background/literature review/gap of knowledge/objectives, etc. The same applies to the other sections: methods, results, and discussion.

Response: Thank you. We have introduced subheadings to each section as instructed.

Line 106: was the search from inception to 04-07-2023? Please mention.

Response: We have added the exact timeline of our search (Line 115)

Line 177: Can you provide references for these meta-analyses? Perhaps you could discuss these with your findings in the discussion section.

Response: We added a reference in our supplementary S10 File- Detailed search strategy (Line 136). Also, we have added the reference in the discussion (Line 491)

Line 118: I suggest this part should go into a subheading, such as screening eligible studies. 

Response: We have added a subheading as instructed (Line 127)

Please mention the initials of each author responsible for reviewing. Generally, conflicts between the primary and second reviewers are resolved by a third reviewer, if required.

Response: We have added the initials of the authors (Line 128-133)

Please add a sub-title explaining all the categories: familiar versus unfamiliar, trained versus untrained, and what temporal association is.

Response: We have added a new sub-title and explained all the categories (Line 168-176)

How did you divide the geographical territory? WHO has a categorization.

Response: Thank you. We used a continental basis to divide geographical territory. We have justified with references (Line 176)

Line 130: add 'viable' singleton pregnancies

Response: We added 'viable' (Line 141)

Line 130: early labour, more accurately, the latent phase.

Response: We agree. Added 'the latent phase' (Line 141)

Line 132: please explain what you mean by genital abnormalities here

Response: We acknowledge that we have mistakenly mentioned genital abnormalities. We have changed it to 'pelvic abnormalities not favor vaginal birth' (Line 144)

Lines 133-134: What about quasi-experimental studies? Otherwise, you can simply mention that any other study designs, other than randomised controlled trials, including quasi-experimental trials, were excluded. Did you have limitations for the article language? Were all the articles in English?

Response: We have changed as instructed (Lines 145-146 and 143)

Line 135: Suggest not mixing the study outcomes with the data extraction and quality assessment. Study outcomes should be a separate subtopic followed by another subtopic for the 'Risk of bias and quality assessment'- RoB2 and possible biases. In the study outcomes, please mention primary and secondary outcomes clearly.

Response: Thank you. We have amended the article to separately mention primary and secondary study outcomes (Line 158-166). We introduced a separate subtopic, risk of bias and quality assessment, as advised (Line 152-157). 

Line 144-146: Risk of bias assessment – I am not sure the authors have used the current RoB -2 tool as the domains that appear on the manuscript are different from the RoB -2. Looks like authors have used RoB -01. Please update accordingly.

Please go to this link for more information on RoB 2: https://sites.google.com/site/riskofbiastool/welcome/rob-2-0-

tool/current-version-of-rob-2

Response: We acknowledge that we have used ROB 01 for the risk of bias assessment but not ROB 02. We amended the article to reflect the facts (Line 153-155)

Line 150-160: Suggest using 'versus' instead of 'vs'.

Response: We changed accordingly (Line 186-188)

Table 1: Please provide what LSCS means as a footnote. All tables/figures should be self-explanatory. 

Response: We changed LSCS to CS throughout the article. We added a footnote (Line 247)

Can not use RCT in the table legend. Please word it out.

Response: Thank you. We have changed as instructed (Line 245)

Table 1: What is the order of appearance of the RCTs in Table 1? It should be either the chronological order of appearance, A-Z or any other rational categorization.

Response: We re-organized according to alphabetical order.

Table 1, 6th column: what are these outcomes? Are these the primary outcomes reported by the trials? Or else all the outcomes? It would be more explanatory to divide primary and secondary outcomes.

Response: These are the primary outcomes reported by the studies. We changed the column heading accordingly.

Line 219: There cannot be more than one funnel plot to assess the publication bias for one meta-analysis. Could you please elaborate more?

Response: We had eight funnel plots for each forest plot drawn to analyze primary outcomes (Line 248)

In Tables 2/3/4, I2 for heterogeneity appears as a decimal value. Is this a mistake? According to the Cochrane Handbook, I2 should be 0-100% as a percentage.

Response: Thank you. We agree. We have changed the I2 decimal to % value throughout the manuscript. 

Line 260: cesarean section, which is abbreviated as LSCS. Please correct.

Response: We removed the abbreviation 

Suggest American English terms like labor, not 'labour' throughout the manuscript.

Response: We changed to 'labor' throughout the manuscript.

Table 4: 6th column – please explain what you mean by the test for subgroup difference.

Response: Statistically significant subgroup difference is trial / patient characteristic/covariate considered in the subgroup analysis modifies the outcome significantly. P value from the test for subgroup difference is used to assess statistically significant subgroup differences. Usually, P<0.1 indicates a statistically significant subgroup difference (Line 273-276)

Discussion:

The first paragraph needs to be rewritten for clarity. It should start with the research gap and objective and end with the main findings.

Response – Thank you for the guidance. The first paragraph was rewritten to reflect the suggested outline (lines 328 - 342). The second paragraph summarizes the main results (343 – 361). 

Please divide the entire discussion into separate sections as sub-topics to increase the readability of the paper. Please follow the steps of a well-written systematic review publication in a reputed journal. For example: Main findings, Critical review of literature, Strengths and limitations, Conclusions, Recommendations, Future directions etc.

Response – Thank you. We have subdivided the discussion into subheadings as suggested.

Line 312: Define ICU/SCBU abbreviations

Response – expanded into words (Lines 376-377)

Reference list:

None of the references have included the journal name.

Response: We have added the journal name to each reference.

We thank both reviewers for appreciating the scientific merit of our submission and encouraging comments made. We sincerely appreciate both reviewers for their time and expertise granted in the comments suggesting vital improvements toward a publication. We have made the necessary amendments and hope the article now meets the expectations for publication.

---

## [Decision Letter · Decision Letter 1]

18 Jun 2024

PONE-D-24-02940R1Impact of continuous laborcompanion- who is the best: A systematic review and meta-analysis of randomized controlled trials.PLOS ONE

Dear Dr. Jayasundara,

Thank you for submitting your manuscript to PLOS ONE. After careful consideration, we feel that it has merit but does not fully meet PLOS ONE’s publication criteria as it currently stands. Therefore, we invite you to submit a revised version of the manuscript that addresses the points raised during the review process.

The reviewers have recommended minor revisions to your article. Kindly address these before a final decision can be made.

We look forward to receiving your revised manuscript.

Kind regards,

Tanya Doherty, PhD

Academic Editor

PLOS ONE

Journal Requirements:

Reviewers' comments:

Reviewer's Responses to Questions

**Comments to the Author**

1. If the authors have adequately addressed your comments raised in a previous round of review and you feel that this manuscript is now acceptable for publication, you may indicate that here to bypass the “Comments to the Author” section, enter your conflict of interest statement in the “Confidential to Editor” section, and submit your "Accept" recommendation.

Reviewer #1: All comments have been addressed

Reviewer #2: All comments have been addressed

2. Is the manuscript technically sound, and do the data support the conclusions?

Reviewer #1: Yes

Reviewer #2: Yes

3. Has the statistical analysis been performed appropriately and rigorously? 

Reviewer #1: I Don't Know

Reviewer #2: Yes

4. Have the authors made all data underlying the findings in their manuscript fully available?

Reviewer #1: Yes

Reviewer #2: Yes

5. Is the manuscript presented in an intelligible fashion and written in standard English?

Reviewer #1: Yes

Reviewer #2: Yes

6. Review Comments to the Author

Reviewer #1: Thank you for the opportunity to review the revised version of the article. I think that the discussion section is much improved and that the article is ready for publication.

One small suggestion to potentially consider is to perhaps unravel some of the complexity relating to the issue of training and familiarity in the paragraph: 'Training of labor companions versus making trained labor companions familiar'. I thought it might be helpful to state explicitly that we aren't sure if the importance of familiarity is much more significant than the training labour companions receive and that familiar companions tend not to have received any training or whether the training being provided is somehow reducing the effectiveness of labour companions who would otherwise be more effective? And the more work needs to be done to understand this relationship?

Reviewer #2: Thank you for addressing the reviewer's comments. The manuscript is now very much improved and almost suitable for publication in this journal. However, I would like to suggest a few points. Once these have been addressed, it should be suitable for publication following the editorial review.

- Line 310: I2 is expressed as 0.44. Please check.

- Please define all abbreviations used in all tables and figures (such as RR, CI, RCT) at the bottom as footnotes.

- Line 305: You may discuss this finding (length of labor) in the discussion section, probably due to the probable rise of cesarean section rates after 2000.

- Another limitation is the methodological quality of the randomized trials before 2000. There was a huge improvement after 2000, especially with the introduction of trial registries in 2006. These aspects are now being discussed everywhere.

- I noticed a high risk of bias due to blinding: suggest a sentence in the limitations section/discussion section, as blinding is not possible due to the nature of the intervention concerned.

- Please acknowledge the use of the older version - 'original version of the Cochrane risk of bias tool', rather than the currently accepted RoB-02 tool. This is an important limitation. You may also replace the word 'RoB - 01' with the name of 'original version of the Cochrane risk of bias tool'.

- Finally, if the authors are happy (optional request), they can attach the Excel files/summary data files showing the results of the data extraction used to calculate the relative risks for at least the primary outcomes. These can be added as supplementary files. This might enhance the quality of your work.

Congratulations on the good work.

Thank you

7. PLOS authors have the option to publish the peer review history of their article (what does this mean?). If published, this will include your full peer review and any attached files.

Reviewer #1: No

Reviewer #2: No

---

## [Author Response · Author response to Decision Letter 1]

30 Jun 2024

Comments to the Author

Reviewer #1: Thank you for the opportunity to review the revised version of the article. I think that the discussion section is much improved and that the article is ready for publication. 

Thank you for the encouraging comments

Reviewer #1: One small suggestion to potentially consider is to perhaps unravel some of the complexity relating to the issue of training and familiarity in the paragraph: 'Training of labor companions versus making trained labor companions familiar'. I thought it might be helpful to state explicitly that we aren't sure if the importance of familiarity is much more significant than the training labour companions receive and that familiar companions tend not to have received any training or whether the training being provided is somehow reducing the effectiveness of labour companions who would otherwise be more effective? And the more work needs to be done to understand this relationship?

We thank the reviewer for the encouraging comments and agree to add clarifications and comments. We have included a brief discussion as suggested. (Lines 428-455)

Reviewer #2: Thank you for addressing the reviewer's comments. The manuscript is now very much improved and almost suitable for publication in this journal. However, I would like to suggest a few points. Once these have been addressed, it should be suitable for publication following the editorial review.

- Line 310: I2 is expressed as 0.44. Please check.

Response: We thank the reviewer for the encouraging evaluation and the suggestions for improvement. We have corrected it as a percentage. (Line 311)

- Please define all abbreviations used in all tables and figures (such as RR, CI, RCT) at the bottom

as footnotes.

Thank you. We have defined it as suggested in the included footnotes. 

- Line 305: You may discuss this finding (length of labor) in the discussion section, probably due

to the probable rise of cesarean section rates after 2000.

Thank you. We agree. We have included the discussion (Lines 476-480)

- Another limitation is the methodological quality of the randomized trials before 2000. There

was a huge improvement after 2000, especially with the introduction of trial registries in 2006.

These aspects are now being discussed everywhere.

Thank you. We have included the discussion point (Lines 476-480)

- I noticed a high risk of bias due to blinding: suggest a sentence in the limitations section/

discussion section, as blinding is not possible due to the nature of the intervention concerned.

Thank you. We have included a sentence acknowledging the limitation (Lines 518-521)

- Please acknowledge the use of the older version - 'original version of the Cochrane risk of bias

tool’, rather than the currently accepted RoB-02 tool. This is an important limitation. You may also replace the word 'RoB - 01' with the name of 'original version of the Cochrane risk of bias tool’.

Thank you. We have acknowledged the limitation (Lines 30, 154,156, 250, 520-521)

- Finally, if the authors are happy (optional request), they can attach the Excel files/summary data

files showing the results of the data extraction used to calculate the relative risks for at least the primary outcomes. These can be added as supplementary files. This might enhance the quality of your work.

Response: We are happy to share summary data showing the data extraction results. We have attached summary data as supplementary file 12 (S12 File).

Congratulations on the good work.

Thank you.

---

## [Editor Report · Decision Letter 2]

9 Jul 2024

Impact of continuous laborcompanion- who is the best: A systematic review and meta-analysis of randomized controlled trials.

PONE-D-24-02940R2

Dear Dr. Jayasundara,

We’re pleased to inform you that your manuscript has been judged scientifically suitable for publication and will be formally accepted for publication once it meets all outstanding technical requirements.

Kind regards,

Tanya Doherty, PhD

Academic Editor

PLOS ONE
---

## [Editor Report · Acceptance letter]

12 Jul 2024

PONE-D-24-02940R2 

PLOS ONE

Dear Dr. Jayasundara, 

I'm pleased to inform you that your manuscript has been deemed suitable for publication in PLOS ONE. Congratulations! Your manuscript is now being handed over to our production team.

Kind regards, 

on behalf of

Professor Tanya Doherty 

Academic Editor

PLOS ONE